# Understanding Digital Identity during the Pandemic: An Investigation of Two Chinese Spanish Teachers

**Shikun Li** [1], **Junjie Gavin Wu** [2], **Jing Bian** [1], **Zhishuo Ding** [1] and **Yuliang Sun** [3,*]

1   School of Foreign Languages and Cultures, Nanjing Normal University, Nanjing 210098, China
2   School of Foreign Languages, Shenzhen Technology University, Shenzhen 518118, China
3   School of Foreign Languages, Soochow University, Suzhou 215006, China
*   Correspondence: syl870815@gmail.com

**Abstract:** During combatting the COVID-19 pandemic, the most widespread change in Spanish as a foreign language instruction is imperative online teaching. It demands that language teachers move all teaching activities to virtual platforms, facilitating the construction of their digital identities. However, there is scarce attention on Spanish teachers' professional development, given the necessity of understanding the evolvement of their identities across virtual learning platforms. Through the lens of a case study, this research explores the digital identities of Spanish as a foreign language teachers during the school lockdown in 2022. The data includes semi-structured interviews, virtual classroom discourse, lesson plans, and reflective writing. The results show that Spanish teachers formed multiple digital identities, including curriculum innovators, vulnerable actors, involuntary team workers, overseas returnees, and academic researchers. Among them, the first three are core identities, while overseas returnees and academic researchers are peripheral identities. Regardless, they were formed and negotiated under the influence of teachers' past experiences, the exercise of agency, emotional vulnerability, and social context. In addition, a contradictory belief in teaching was also identified during the formation of Chinese Spanish teachers' digital identities.

**Keywords:** digital identity; case study; Spanish language teacher

## 1. Introduction

To effectively combat the COVID-19 pandemic, school lockdowns became the primary choice of Chinese universities in 2022. It demanded teachers move all the learning activities to virtual platforms, which facilitated the transformation from a classroom teacher to a digital teacher. Teachers exercised their beliefs and knowledge of online teaching during the process, which enriched their understanding of education technology, and learners' attitudes toward remote learning. However, the struggle of teachers which were reported during the outbreak of COVID-19 pandemic [1] remains in the current virtual classrooms.

Within two years of intermittent online teaching, teachers had brought their personal background to "serve as a critical component of the classroom sociocultural landscape" [2–5]. Alternatively speaking, as teachers were negotiating personal backgrounds with the needs generated in the new learning contexts every minute, there is a necessity to perceive their online exercises as a series of complicated, ever-changing social activities.

The diverse paths that teachers took to respond to the complicated virtual learning context reflecting and fighting with the hegemonic discourse of the institutions drew our research attention. Teachers formed their digital identities online. In our study, we define digital teacher identity as teachers' knowledge of themselves generated from their interaction with particular technologies and people involved in the virtual context [6]. As the manifestations of teachers' professional identities, digital identity influences lesson delivery, classroom discourse, feedback type, and assessment choice [7–11].

In addition, overlap exists between the construction of teachers' professional identity and their digital identity as well. In a broad sense, teachers form their identities through

"learning a multitude of skills, gaining a breadth of knowledge, and moving from novice to experienced practitioners" [12]. The formation of language teachers' digital identities, likewise, is a dynamic, complicated process in which teachers strive to interpret their "beliefs, values, and education experiences in light of new context and frames of relationship in the contemporary digital society" [13,14]; as cited in [15].

Even though research attempts were initiated to understand teacher identity over the decades [16–20], the depiction of teachers' digital identity and the formation process remains inadequate. Besides, most research results were drawn by studying teachers in English-speaking countries. Few pieces of research has examined the digital identity of language teachers in a foreign context, let alone teachers in the pandemic. The new identity teachers formed online would not disappear, even after the school lockdowns ended. On the contrary, it would influence teachers' professional development through their teaching journey.

In this study, we chose to study Spanish as a foreign language (SFL) teachers rather than English as a foreign language (EFL) teachers for the following reasons: Spanish is receiving growing attention among Chinese higher education institutions, as it contributes to Chinese global strategic initiatives (e.g., the Belt and Road initiative). Recently, as the Spanish programs expanded exponentially, the number of Spanish teachers raises correspondingly in China. Yet, the existing studies on Spanish language teachers' digital identity are still limited, though there is a need to understand their professional development during the pandemic. An insufficient number of studies usually results in a shortage of practical pedagogical suggestions on SFL teacher preparation.

To fulfill the research gap mentioned above, this study explores the digital identities of SFL teachers and their formation process during the school lockdown in 2022. We aim to capture SFL teachers' dynamic digital identities in practice. Our findings are intended to assist teacher educators in comprehending SFL teachers in virtual classrooms.

## 2. Theoretical Framework

As post-structuralism advocates, we see identity as an ever-changing product of social activities containing the symbols of a particular culture and time [2,21–23]. We believe its multiplicity, fragmentation, and conflicting nature is not only manifested through classroom discourse and teaching practices but also is identified by studying teachers' teaching stories [2,24].

### 2.1. Formation of Teachers' Professional Identity

Three characteristics regarding the formation of teachers' professional identities are proposed in a systemic review of the literature by Beijaard Meijer, and Verloop [25]. To begin with, the construction of professional identity is an ongoing process in which teachers attempt to make sense of themselves and their previous teaching experiences while being constrained by the current expectations of others [25,26]. This mechanism shows that though individual differences, such as personality matters, a teacher's professional identity is the consequence of a negotiation between the individual and others in a particular social context.

The second characteristic suggested by Beijaard and colleagues is the emergence of sub-identities while forming one's professional identity [25]. Alternatively, a teacher's professional identity is not unitary but composed of a core and several peripheral sub-identities that are well balanced. Core sub-identities do not readily change in the face of education reform or teaching conflicts. However, peripheral ones are different, as it is less costly to modify them [25]. The last characteristic is that forming a professional identity is an active process in which teachers continue to exercise agency, despite being affected by implicit factors, such as popular beliefs in higher education, or national policies on foreign language learning. Usually, a teacher's agency is initiated by acquiring new skills or collaborating with others [25,27].

In short, scholars believe that the formation of teachers' professional identity is "a process of practical knowledge-building characterized by ongoing integration of what is individually and collectively seen as relevant to teaching" [25].

### 2.2. Digital Identity as a Manifestation of Professional Identity

The digital identity of teachers, regardless of subject, is commonly explored by evaluating the individual's participation in training on education technology or online teaching practices, such as using digital portfolios as assessment, digital storytelling, developing online projects, and social media posts [7,10,15]. Different digital identities have been proposed based on discourse analysis results or qualitative evidence. For example, Yuan and Liu trace how three Chinese English as foreign language (EFL) teachers coped with the abrupt changes in the pandemic (e.g., mandatory online teaching in 2020) and formed their digital identities accordingly. Yuan and his colleague propose that EFL teachers translate their imaged identities into practiced identities during online learning [1]. The imaged identities were formed based on the evidence of best practices in traditional classrooms, and they displayed a decisive influence on teachers' practiced identities. However, due to the restraints of reality, the imaged identities had yielded to online teaching practices, and teachers' digital identities as practitioners were formed correspondingly [1].

Another study investigating the teachers during the COVID-19 pandemic focuses on the faculties in Pakistani universities. By employing a mixed method design, Shahnaz presents the challenges in online teaching and the additional support (e.g., IT-support, flexible working schedule, and workshops on effective online education, etc.) provided by some domestic universities. According to the author, technical or professional support from universities was not expected during the pandemic and was rarer in the past [28]. Teachers who received less institutional assistance would choose to either self-teaching or consult with students with knowledge of the internet [28]. However, despite having poor digital literacy, Pakistani teachers formed an "optimistic, empathetic and power-sharing with learners" identity online [28].

In her year-long ethnographic case study, Curwood concludes that teachers who attended workshops on integrating technology with lesson plans used verbal and semiotic resources to construct their identities [7]. Some teachers perceived that technology integration slowed down the process of establishing an "ideal teacher", whereas others saw learning how to use technology as part of social learning, leading to a positive attitude toward digital tools [7]. Behind these conflicting, verbal, and non-verbal cues were the characteristics of teachers' digital identities.

Likewise, Engeness evaluates how teachers form digital identities by attending workshops adopting Galperin's pedagogical principles in designing and implementing online courses [15]. Through the lens of two online teaching scenarios, Engeness concludes that her participants could select appropriate education technologies to facilitate online instruction, and correspondingly they formed digital identities as "conscientious users and co-designer of digital environment" [15]. Teachers' engagement in online learning not only verifies their capacity to exercise agency but also contributes to an evaluation of their educational beliefs and knowledge of how to become digitally competent teachers [15].

In addition to professional workshop evaluations, the analysis of teachers' discourse is also seen in the study of digital identities [10]. Honan uses rhizotextual analysis to relate language art teachers' discussions about education technologies with the policy discourse (e.g., the rhetorical use of language in official documents) [10]. In her study, Honan refutes perceiving teachers as compliant with educational policies, but sees them as individuals who formed digital identities as experts using professional terms to interrogate the integration of digital text in literacy teaching [10].

Another associated study argues that although teachers may be literate in digital texts across different virtual platforms, they like to consult similar digital resources to maintain their professional identity [29]. These homogeneous selections are not a coincidence but a reflection of pre-service teachers' belief in "appropriateness" rooted in a particular learning

context [29]. Based on three pre-service teachers' digital practices, Burnett claims that digital literacy practices do not work as resources, but as the basis of one's digital identity. Furthermore, Burnett argues that even though being influenced by "appropriateness", pre-service teachers could construct individualized, context-specific digital identities with reflective self-narratives [29].

Not only virtual teaching practices, but also social networks, influence teachers' digital identity formation [9,30]. In Fox and Bird's study, as the qualitative evidence yields, there is tension between teachers' personal and professional virtual identities [9]. However, instead of emphasizing the contradictions, the authors propose that the process developed "synergically", indicating that the effects of tension are not entirely adverse. For example, a teacher who successfully merged his online and offline identities felt more control over his professional and personal life [9]. Moreover, teachers' social network engagement has strengthened their competencies in sharing information and managing multiple digital identities across online learning platforms [30].

Even so, some teachers suffer from societal pressure to become active members of the online community, which would result in reduced teaching efficiency. Though Fox and Bird's study does not focus entirely on the characteristics of the participants' digital identities, it raises researchers' awareness of the role of social networks in constructing teachers' identities [9].

An earlier study on the relationship between social networking and digital identity generates a similar but broader conclusion: social networks are stages where people can exercise their digital presence in multiple ways, consciously or unconsciously [31]. Regardless of the type of presence, a digital identity reflects the dominant personality instead of balancing the "ideal" self or a "real" self [31]. Meanwhile, formed in the virtual ecosystem, peoples' digital identities are mediated by interpersonal communication, catalyzed by virtual social networking, as it improves the intimacy and immediacy of communication [31]. Though the study did not stress teachers' digital identities, its explanation has deepened our understanding of digital identity formation. The studies mentioned above provide an overview of teachers' digital identities, as well as the factors that impact their formation. Obviously, some of the conclusions echo the theory regarding teachers' professional identities proposed by Beijaard and colleagues, while the others call for further exploration of the paths of digital identity formation, especially during the lockdown. Therefore, to fill this research gap and compensate for the lack of attention to the professional development of SFL teachers, we posed two research questions:

1.　What are the characteristics of the digital identities of two Chinese Spanish teachers during the school lockdown?
2.　How did these two Chinese Spanish teachers form their digital identities?

## 3. Method

To better capture the dynamics of Chinese SFL teachers' digital identity, we employed the case study method. According to Yin, a case study is more effective when exploring contemporary social events where the boundary between the phenomenon and the social context is unclear [32].

### 3.1. Participants

This study recruited two SFL teachers from a public university in southeastern China through emails and phone calls. Our participants are native speakers of Chinese and both obtained their Ph.D. in Spain, but from different academic programs. The male Spanish teacher, William Yang (a pseudonym), who is 34 years old, majored in applied linguistics with a research concentration on syntactic structure, whereas the female teacher, Selina Zhi (a pseudonym), 32 years old, studied translation theory. During the semi-structured interview, they were invited to identify their Spanish proficiency in four language skills. Both SFL teachers reported that their proficiency in Spanish speaking, listening, and writing was at the C1 level. Reading was at the C2 level, according to the Common European

Framework of Reference for Language (CEFR). C1 and C2 level Spanish learners fall in the general category of "proficient user" [33]. Specifically, as CEFR suggests, our participants have the linguistic competence listed in Table 1 [34].

**Table 1.** Description of proficiency levels for SFL teachers [34].

| Proficiency Level | Description |
|---|---|
| C2 Reading | Read with ease virtually all forms of the written language, including abstract, structurally, or linguistically complex texts such as manuals, specialized articles, and literary works. |
| C1 Speaking | Understand extended speech even when it is not clearly structured and when relationships are only implied and not signaled explicitly. Understand television programs and films without too much effort. |
| C1 Writing | Express themselves in clear, well-structured text, expressing points of view at some length. Write about complex subjects in a letter, an essay, or a report, underling what is personally considered salient issues. Select a style appropriate to the reader in mind. |
| C1 Listening | Present clear, detailed descriptions of complex subjects, integrating sub-themes, developing particular points, and rounding off with an appropriate conclusion. Express themselves fluently and spontaneously without much obvious searching for expressions. Use language flexibly and effectively for social and professional purposes. |

During the school lockdown in 2022, these two participants co-taught an undergraduate course, Fundamental Spanish I, online. Although the course was virtual, they developed the lesson plans collaboratively.

### 3.2. Context

This study was conducted in a public university located in eastern China. Unlike many Chinese public universities, the Spanish department at the research site offers a dual degree rather than a bachelor's degree in Spanish. Undergraduate students are provided with two options: a English-Spanish major or a Spanish-English major. As the name suggests, the English-Spanish major undergraduates have to register for more academic credits in English language arts, whereas Spanish-English majors need to complete more coursework in Spanish literature and related fields. The core courses do not start until the second year of undergraduate study, and prior to that, regardless of the major, freshmen in the Spanish program are offered the same courses. An individual's major is determined by the grades of finals, along with the learner's desire at the end of the freshman year. The participants in our study are teachers from the Spanish-English major program.

The average class size of the Spanish–English major program is 20 to 25 students and, depending on the level of the course, the department sometimes assigns two teachers to co-teach one class over the course of a semester to fulfill the requirements of teaching hours set by the university, as the department was not able to open enough classes. Our two participants were co-teaching a course named Fundamental Spanish I.

### 3.3. Data Source

**Semi-structured interviews**. Three semi-structured interviews were conducted with each SFL teacher, aimed at understanding their digital identities and personal background. The purpose of the first interview was to become acquainted with the two SFL teachers and obtain as much information as possible about their prior experience. Questions in the first interview were about their educational background, professional training, and virtual learning while attending graduate school. The second semi-structured interview focused on elaborating on the practices of online teaching. Questions of the second interview inquired

about the SFL teachers' beliefs and knowledge of online teaching, their digital literacy, teaching innovations, significant online teaching events, teacher–learner relationships, and, most importantly, their visions of their "ideal" self during online instruction. The last interview underlined the development of the SFL teachers' mindset, the impact of their sub-identities on digital identity formations, and the emotion fluctuations during online instruction. All interviews were conducted in Chinese to ensure the clarity of the questions.

**Recording of online classes**. This study included recordings of six online lessons extracted from the Fundamental Spanish I course, which is a core course for sophomores in the Spanish–English dual major, with an emphasis on introducing Spanish grammar principles, phonetic symbols, and vocabulary, as well as essential reading and essay-writing techniques. The six lessons revolve around instruction on one element of Spanish grammar, aspect. These video recordings allowed us to comprehend how the SFL teachers practiced their beliefs and pedagogical knowledge while teaching online, as well as the power relationship in the virtual classroom.

**Other artifacts**. Apart from semi-structured interviews and online classes, we also collected other teaching artifacts, such as the lesson plans and reflections on teaching each class, and a recording of an online meeting between the two participants in which they planned one of the six lessons collaboratively. These artifacts helped evaluate the differences between the delivery of the lessons and the teachers' design of the online course.

**Researcher's field notes**. Field notes were composed as the researchers watched the recordings of online classes. In these notes, the researchers described the power relationship between the learners and their teachers, the feedback circle, and the pace of teaching.

**Survey**. A Chinese Spanish Learners' Perceptions of the Teachers during Online Instruction Survey included items graded on a 7-point Likert scale (1 = completely disagree, 4 = neither disagree nor agree, and 7 = completely agree). It was administered after the spring semester of 2022 to boost the validity of the results. The survey contained 12 items, with 9 of them designed to assess the learners' perception of the teachers' performance online and the rest to inquire about the professionalism of SFL teachers during online instruction.

### 3.4. Data Collection and Analysis

Data were collected over the course of 10 months, which included both the spring and fall semesters of 2022. The data collection started with the first semi-structured interview at the beginning of the spring semester (13 February 2022), followed by the recording of the online classes, collection of other teaching artifacts, and field notes. Close to the end of the semester (30 June 2022), the second semi-structured interview was conducted. The last interview was done on 5 December 2022, to track the changes in the SFL teachers' mindset. All qualitative data were translated and transcribed by the same researcher and, to improve content validation, all transcripts were proofread by the participants afterward.

Due to the paucity of literature on SFL teachers' digital identity, we chose the ground theory rather than an existing framework. Two rounds of coding were carried out to decode the digital identities. The first round of line-by-line coding allowed us to tune in to the participants' perspectives and ruminate on the questions in the semi-structured interviews, online classes, and lesson plans. The second round of focused coding helped us in clustering codes, as well as identifying units of meaning. Themes regarding the characteristics of Spanish teachers' digital identity and its formation were also generated in the last coding round.

In addition to qualitative data, this study also used a survey to obtain feedback on SFL teachers' online practices, and it was administered to learners who had enrolled in the Fundamental Spanish I course in the Spring semester of 2022 that was co-taught by Dr. Yang and Dr. Zhi. In all, 19 junior Spanish-English majors voluntarily completed the survey, and their responses were analyzed via SPSS 35.

## 4. Results

The two SFL teachers in this study have different personalities, years of teaching experience, and paths of professional development. However, our data suggested that their digital identities possess some similar characteristics.

### 4.1. New Platform, New Teachers

The imperative of online teaching during the school lockdown spurred teachers to invent new ways to cope with the changes. During online instruction, our participants rejected the traditional approaches of language instruction and actively implemented teaching innovations. In the semi-structured interview, William described how he used independently designed videos to assist learners with acquiring abstract Spanish grammar principles:

> I would like to assign videos to them before the class, and these videos include the grammar knowledge of the textbook. Then during the online teaching, I would ask questions to assess their comprehension. In this way, if learners failed to answer the question or didn't understand my explanations in the video, they could always re-watch it. (William, the second semi-structured interview)

Apart from videos, William also mentioned the virtual group presentations. This meaningful group activity was aimed at increasing the learners' capacity to explore new concepts introduced in online classes, as well as facilitating peer collaborations. After forming groups voluntarily, they were encouraged to present the key concepts of the textbook during the virtual group presentation, as well as to integrate reflection questions, content-related videos, and supplementary learning resources into the slides. Each virtual presentation was limited to 10 to 15 min and was usually presented at the beginning of the online class.

In the videos of the online classes, the learners seem drawn to this new way of presenting knowledge and actively answering their peers' questions. This was in sharp contrast to their attitude while answering the teachers' questions. "This activity is quite fresh for sophomore Spanish learners, and by doing it, they are able to connect new knowledge with their existing knowledge" William commented.

This teaching strategy was seen as a breakthrough by the Spanish teachers in our study, who had less pedagogical training during their graduate study. William confessed in one interview that he would not include any grammar explanation video or the virtual group presentations in the traditional classroom because it was very time-consuming to develop, and he was constrained by the timeline set by the department.

These teaching innovations eventually contributed to the development of the SFL teachers' digital literacy, and the formation of their digital identities. Digital literacy in this study was defined as teachers' competence in selecting and using different educational technologies to assist language learning, as well as presenting themselves online [30].

Selina's teaching innovation focused on building the learners' confidence in the virtual classroom. When facing their reluctance to respond, Selina rearranged the order of difficulty of questions. Instead of rushing into questions that promoted higher-order thinking skills, she started with simple questions, for example, about practicing pronunciation or translating new words. As the difficulty decreased, learners' motivation increased. Meanwhile, Selina provided active learners with more opportunities to answer questions, with the hope that they would have an influence on the shyer ones. After noticing an improvement in learners' participation, she steadily adjusted the difficulty of the assessments. Selina said, "Online teaching makes me realize that I need to boost learners' confidence first and then consider the participation".

Engaging in online teaching not only gave rise to new teaching ideas, but also inspired Spanish teachers to picture their ideal education technologies. For example, William envisioned the following:

I really wish there was some media resource that matched our textbook, maybe a movie. Say, if I am teaching Spanish aspect, I could use the movie clips to illustrate the changes between the perfect and the imperfect aspect of Spanish. As long as it contains the scenarios I need, and then I struggle no more to find appropriate videos. (William, the second semi-structured interview)

The switch between offline and online teaching presented many challenges, but it also created new identities for language teachers as curriculum innovators. Obviously, Chinese SFL teachers initiated their agency in online teaching by actively addressing the needs of learners, experimenting with new strategies, and, last but not least, envisioning an "alternative way to learn Spanish online".

Lastly, our survey results showed that the learners held a very positive attitude toward the new attempts mentioned above (M = 6.87, SD = 0.46). Specifically, the statistics indicated they graded Dr. Zhi's online performance (M = 6.89, SD = 0.27) slightly higher than Dr. Yang's (M = 6.76, SD = 0.33), suggesting that they tended to agree more with items indicating that Dr. Zhi's online instructions were effective.

### 4.2. Vulnerability of Spanish Teachers during Online Teaching

Both SFL teachers in our study had trouble effectively communicating with their learners remotely; as a result, they attached importance to the capacity to engage learners online. For example, when they were asked to describe a competent Spanish teacher during the semi-structured interview, they included characteristics such as "capable of engaging his learners in online classes". The video of online classes exhibited our participants adjusting their performance according to the learners' verbal or facial feedback. "I feel that I was acting alone", recalled William in one interview. He preferred to use the word "act" to depict online teaching, suggesting the power inclination toward learners in the virtual classroom.

The "one-man show" metaphor also expresses the teachers' complaints about language learners' passivity, exemplified by comments such as "Most of the time, they would not open the camera, and even if they were required to, you still did not know what they are doing" (William, the second semi-structured interview) or "I doubt they might have the textbook open in front of them when answering the questions", (Selina, the second semi-structured interview) which were signs of teacher fatigue. Selina likened some of her teaching practices online to "one punch to a sponge". Even though the silence of learners could be due to many causes, such as poor motivation, shyness, or not wanting to stand out among peers, it significantly affected our participants' perception of online classes.

However, although these SFL teachers showed emotional vulnerability during online instruction, the survey results showed that the learners' perceptions of their professionalism were not affected (M = 6.73, SD = 0.33).

### 4.3. "I Prefer to Work Alone"

Since online teaching began, William and Selina have had several successful collaborative attempts, such as design assessment, lesson planning, and discussing lesson delivery. They did express gratitude towards the ideas and help obtained from their partner. However, in the last interview, neither said they wished to continue collaborating. Despite the many challenges mentioned earlier, the SFL teachers in our study would rather deal with them independently in the future. Unexpectedly, the reason they provided was the same: "Collaborative teaching costs too much time. I would like to invest more energy on research and get my writings published, as it was the requirement of getting tenure" (William and Selina, the second semi-structured interview).

Apparently, none of them perceived online teaching as at the top of their to-do list, and the positive experience of collaborative teaching remained inadequate to change their priorities.

*4.4. Digital Identity Is Formed over the Course of One's Teaching Career*

Rather than claiming that SFL teachers formed their digital identity exclusively through online teaching, we propose that one's digital identity evolves from one's professional identity. By breaking down the teaching journey into separate stages, we reveal how digital identity emerges from one's previous teaching experiences.

***William's Journey of Forming a Digital Identity.*** William's first teaching task started in the last year of his undergraduate study when the Confucius Institution recruited him to teach Chinese in Chile. Without a thorough learning of pedagogical principles, William was assigned to teach Chinese in a class where foreign language learners had diverse learning needs, mixed proficiency levels, and varied motivations. He reflected that teaching strategies were obtained mainly through informal communication with experienced teachers and observing other teachers' practices.

The one-year teaching assignment seemed to contribute little to William's formation of a professional identity, as he constantly referred to the experience as "interest-oriented classes" and reported challenges in maintaining course progress. However, though entirely new to language teaching, he was aware that students' dropping out or being absent without an excuse was a consequence of frustration caused by pronunciation problems or poor motivation. He applied these acute observations to online teaching later, which cast an influence on his teaching innovations.

During his first teaching assignment, William had no experience teaching or attending online classes, but he thought of using videos to display cultural elements. William recalled that videos had successfully improved his teaching efficiency as they assisted learners in comprehending and practicing cultural knowledge. Back then, he had barely seen any modern education technologies, such as Smartboards or iPads, which might have slowed the development of his digital literacy.

After one year of teaching overseas, William returned to China and entered the second stage of his teaching career. During this stage, he worked as a Spanish teacher at a domestic college. We identified the initial formation of his professional identity through the narration of his teaching experiences there. He said:

> I am more nervous after returning to China and becoming a Spanish teacher at a local college. Much more than the first time in Chile. I need to be more focused and more professional in a Chinese college. . . . I am officially a Spanish teacher now, and my learners would challenge me more. (William, the second semi-structured interview)

Obviously, for William, the teaching tasks at the college level were more diverse and complicated. His professional identity as a Spanish teacher has been constructed accordingly. However, William's digital literacy remained the same through the first two teaching stages: online videos were the primary tool, whether for introducing abstract cultural phenomena or being used as a supplementary resource to enhance Spanish language learners' listening skills. Prior to 2020, William had not received any systematic training on how to deliver online courses; instead, he gained some experience by attending open-access courses online. In the interview, he spoke less about the impact of these online videos, as he believed those classes "apparently were created by a professional team, and more importantly, it was not language class".

The widespread lockdown during the COVID-19 pandemic catalyzed the formation of William's digital identities because nearly every teacher in his university was demanded to implement courses online. Based on William's narrations, we identify that he formed his digital identity by ruminating on previous virtual teaching practices, such as implementing innovative teaching strategies, effectively addressing learners' feedback online, collaborating with colleagues, and, very importantly, ensuring his online teaching ideas conformed with school policies.

Meanwhile, we also notice that William had formed two sub-identities as an applied linguistics researcher and an overseas returnee. William said his previous training in

academic research contributed to his success in implementing new teaching ideas because he knew how to test them with a small number of learners before applying them in a larger online class.

William recalled that his sub-identity as an overseas returnee was manifested through sharing cultural facts with his learners online. He said: "My identity as an overseas returnee allows me to discuss some uncommon but fun cultural facts during online instruction".

The sub-identities of William were intertwined, and he prioritized displaying each or a combination of several based on the situations. Over the course of the Spring semester in 2022, William's digital identity as a language teacher was the most dominant identity, while at online academic conferences, his identity as a researcher in applied linguistics would be more apparent.

*Selina's Journey of Forming a Digital Identity.* Selina obtained her master's and doctoral degrees in Spain, as William did, but she did not wish to become a language teacher in the first place. "I majored in Spanish and finance, and I did not expect to become a language teacher", Selina recalled during the first interview. After obtaining her M.A. in Spanish, Selina returned to China and started her first teaching assignment at a public university near her hometown, where she fell in love with teaching and decided to pursue an advanced degree in Spanish-Chinese translation. Unlike William, Selina was not a full-time student in the doctoral program; instead, she had to co-teach some classes with her colleagues in China. Because of the heavy teaching duties, Selina could not work as an international teaching assistant and therefore did not gain any formal overseas teaching experience. Regardless, Selina had learned some effective teaching practices from her teachers in Spain. She recalled:

> One of the impressive teaching strategies (of my teachers in the doctoral program) was that teachers did not have a fixed answer to a question. They encourage learners to justify their answers. I am amazed by this. Because I was an excellent Chinese student in the traditional sense, I was trained to remember the "right answers" but not given any space for rumination and exploration. Through watching my professor's method of teaching, I was inspired. (Selina, the second semi-structured interview)

The "no fixed answer" strategy was later integrated into Selina's online teaching. This agrees with earlier research that pointed out teachers benefitted from the "apprenticeship of observations" [35,36] when they entered a new phase of teaching and, in our case, mandatory online teaching during the school lockdown.

Although she was assigned fewer teaching assignments than William, Selina had more informal online teaching experience prior to the school lockdown in 2022. Selina recalled that during her graduate study, she used Skype to teach Chinese to a foreign language learner: "But it was more like face-to-face communication, and you could always obtain instant feedback online", she recalled. Due to the platform and the class setting being too different from her current teaching scenario, Selina thought her previous online teaching had less impact on her future online teaching practices.

Selina's pedagogical knowledge was acquired mainly through watching teaching competitions online or attending domestic workshops on foreign language teaching. These experiences enriched Selina's knowledge in terms of analyzing the traits of Chinese language learners and being more considerate when facing a decline in engagement in the online learning environment.

Resembling William, Selina's sub-identity as an overseas returnee was more apparent while sharing cultural anecdotes or answering questions concerning pop culture. However, her sub-identity as a researcher in translation theory was not very evident in her online teaching. "My identity as a researcher in translation theory might influence my teaching in general, like I would introduce some translation theories to my learners during offline teaching, but not online", she confessed during the last semi-structured interview.

Starting in 2020, engagement in online teaching also facilitated the formation of Selina's digital identity and yielded unexpected changes in her beliefs about online teaching.

Successful attempts, such as elevating learners' motivations to learn Spanish, or creating an effective feedback circle, raised Selina's confidence regarding online teaching. However, despite her online instruction being acknowledged by others, Selina still firmly believed that online education was no more than a backup plan for Spanish learning. This belief reduced her investment in developing digital literacy and probably slowed down the formation of her digital identity.

In short, Selina's digital identity formation journey resembles William's in some ways but informs us more about her personal teaching belief—she was prone to quoting her experience when attempted to manage the problems that occurred in the virtual classroom. Selina said: "I personally enjoyed a relaxed learning environment, and I believed my learners did so as well".

As a person who preferred social distance, Selina stressed that she barely built any friendships with learners, either online or offline, but strived to maintain professionalism. Her serious attitude was well acknowledged by her learners (M = 6.89, SD = 0.27). However, this mindset was in strong contrast to William's. William was apt to see himself as between a teacher and a friend, but it did not affect learners' perception of his professionalism (M = 6.87, SD = 0.46).

## 5. Discussion

By analyzing qualitative and quantitative data, this study explores the digital identities of Chinese SFL teachers during the lockdown in 2022 and how they were formed. The results show that the teachers were apparently unprepared for online teaching when the school lockdown was announced [37]. Such unpreparedness was not exclusive to SFL teachers but was reported in studies on language education during the pandemic worldwide [38,39]. During the school lockdown, SFL teachers endeavored to cope with the changes in the learning environment while experiencing emotional vulnerability. Their digital identities, which included core identities and sub-identities, illustrated the agency of SFL teachers and the negotiation between one's previous experience and the particular social context.

### 5.1. Digital Identities Are Manifestation of Agency with Individual Differences

According to Duff, agency is defined as people's "ability to make choices, take control, self-regulate and thereby pursue their goal as individuals, leading potentially to personal or social transformation" [40]. In our study, the SFL teachers exercised their agency through curriculum innovations. The lesson plans and recordings of online instruction yield that the SFL teachers invested significantly in implementing new learning activities and adjusting teaching based on the learners' feedback. The reflective writings indicate that our participants prioritized learners' needs and regulated themselves accordingly. For instance, William wrote, "I need to pay more attention to learners' mistakes and watch out whether they grasp the key points of aspect mentioned in the video". Phrases such as "worth attention", "pay attention", and "need to be repeated" were constantly found next to detailed descriptions of learners' online reactions in William's teaching reflections. The end-of-semester survey on learners' perceptions of the efficiency of online teaching verified that our participants' efforts were well paid.

However, individual differences matter in terms of exercising teacher agency. For instance, William's learning innovations were based on his concern that learners could not catch up with the instructors when attending streaming courses. Therefore, he created videos catering to the learners' needs for reviewing. These self-developed videos reflected his acute observations of the learners, as well as his empathy towards learners who struggled because of the unfamiliar learning situation. Selina, in contrast, focused more on improving learners' online participations. Her rearranging of questions and reliance on peer influence displayed more about her emotional vulnerability regarding the less motivated learners.

However, the exercise of agency by these SFL teachers was constrained by their digital literacy. The Spanish teachers in our study were not equipped with sufficient knowledge of online teaching and did not receive any systematic training on virtual learning. No technology other than video and screen-sharing during streaming was used for language instruction. Therefore, despite the desire to make breakthroughs online, their teaching practices no doubt had limitations.

### 5.2. Emotional Vulnerability Is an Integral Part of SFL Teachers' Digital Identities

During the interviews, our participants revealed their struggles with obtaining effective feedback from learners online. The phrases "I do not know", "not sure", or "cannot do it" illustrated the challenges they faced when attempting to use contemporary education technologies during online instruction. However, the lack of role models or professional assistance led to the formation of a digital identity as a "vulnerable actor".

The one-man show metaphor highlights the vulnerability of SFL teachers in the online environment. The SFL teachers in our study confessed that they felt exhausted after talking to a screen with a list of names on the sidebar rather than people's faces. The teachers initially assumed that the virtual platform was responsible for their ineffective teaching and felt a power fading in the virtual classroom with learners manifested more control. However, neither of our participants believed that it was necessary to communicate their frustration with the learners. Lasky defined vulnerability as "a complex, multi-dimensional, multi-faceted emotional experience" [41], and she proposed that it contains two dimensions: openness and protectiveness [37,41]. These two dimensions are not fixed, but transformable, and moving between these two dimensions can lead to a change in teaching practices [37,41]. Our participants displayed protective vulnerability when they were anxious about online teaching and attempted to avoid possible ineffective communication with learners.

However, despite confronting power fading and emotional vulnerability, the SFL teachers' agency remained active. In the virtual classrooms, well-developed questions arose to manage the anxiety caused by the "one-man show" and enhance learners' motivation. According to the recordings of online instructions, there were noticeably fewer dull grammar instructions compared to the offline setting. The structure of online classes had become coherent and characterized. We believe, as previous studies suggest, that the professional role of teachers was reinforced through their attempts to figure out themselves in relation to others, as well as based on labels bestowed from the outside [2,42,43].

### 5.3. Forming a Digital Identity in Social Context with Contradictory Belief of Teaching

Previous studies reported that the digital identities of language teachers are situated in a broad social context [25,44,45]. Alternatively, the digital identity of a teacher is the product of a series of social political activities: on one hand, it is influenced by the traditions and hegemonic discourse of the society, and on the other hand, it does not conform to all of them [24,46–48]. In our study, we did not identify evidence that the SFL teachers were fighting against the hegemonic discourse of society but noticed that they held a similar teaching belief about the accent. When inquiring whether they were concerned that their accent was being exaggerated during online teaching, our participants agreed that it was not an issue because "every Chinese Spanish teacher has an accent, and it is ok". This response implied that they constantly compared themselves to the general Spanish teacher population, and as long as it was a common characteristic, they would not bother to change.

Another typical mentality is that our participants displayed no interest in collaborative teaching but firmly agreed they were team workers. It sounds contradictory, but this belief was common among language teachers in China. Xun analyzed a similar contradictory teaching belief while investigating Chinese teachers' professional identities [49]. Xun claimed that Chinese middle school teachers advocated a new policy of quality-oriented education while secretly insisting on using conventional teaching approaches, the critical component of which was learners' grades [49]. There might be a more elusive mechanism

behind the contradictory thinking we identified in this study. Still, if so, it would affect the formation of digital identity by causing a rise of confused online teachers.

## 6. Implications and Research Limitations

The widespread lockdown and mandatory online teaching created a renewed definition of a "qualified teacher". It simultaneously affected language teachers' beliefs, the exercise of agency, and discourse. Even though the new learning environment inspired Spanish teachers to innovate and experiment with new teaching strategies, traditional teaching practices were still widely employed, such as assessing learners' acquisition of new vocabulary through translations or relying significantly on the arrangement of content in the textbook. Therefore, based on Spanish teachers' digital identity and its formation process in this study, we propose that to improve the online teaching efficiency, the primary task is to improve their digital literacy, including by providing workshops on contemporary education technology, the traits of learners in online classes, or multi-modal assessment. Increasing teachers' digital competence would boost their confidence in using technology, and their desire to implement new ideas. In addition, attention should also be given to explaining the association between pedagogical innovation and teachers' professional growth, allowing teachers to be aware of the importance of sharpening their teaching skills to cater to the needs of learners.

Meanwhile, the reluctance to engage in collaborative teaching is not rare among language teachers at higher education institutions because teacher evaluations predominately focus on their academic success rather than their teaching performance. Therefore, only a revolution in teacher evaluation criteria, such as assigning more weight to teaching performance, would bring teaching back to the top of the list of professional development.

At last, the most significant limitation of this research design was the number of cases. Therefore, the results drawn from the data regarding these two teachers cannot be applied to the general population but can only be used as a reference for future studies. In addition, we only used a short survey on learners' perceptions of SFL teachers during online instruction but had in-depth conversations. Semi-structured interviews or focus groups would be more effective with regard to collecting information from learners.

**Author Contributions:** Conceptualization, S.L. and Y.S.; methodology, S.L.; software, S.L.; validation, S.L.; formal analysis, S.L.; investigation, S.L. and J.G.W.; resources, S.L., Y.S. and Z.D.; data curation, J.B. and Z.D.; writing—original draft preparation, S.L.; writing—review and editing, Y.S.; visualization, Y.S.; supervision, S.L.; project administration, Y.S.; funding acquisition, Y.S. All authors have read and agreed to the published version of the manuscript.

**Funding:** This research was funded by the National Social Science Foundation of China, grant number 20CYY001.

**Institutional Review Board Statement:** Not required at our institution.

**Informed Consent Statement:** Oral consent was obtained from all participants.

**Data Availability Statement:** The data are available upon reasonable request.

**Conflicts of Interest:** The authors declare no conflict of interest.

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
