# Peer review of "Understanding Digital Identity during the Pandemic: An Investigation of Two Chinese Spanish Teachers"

_sustainability, doi:10.3390/su15021208_

Round 1
Reviewer 1 Report
1)What is the main question addressed by the research?
The main question addressed by the study is the impact of COVID-19 on the study, and how blended-learning can favor the development of teaching and learning strategies.
2. Do you consider the topic original or relevant in the field? Does it
address a specific gap in the field?
3. What does it add to the subject area compared with other published
material?
The topic, although not genuine, is original, since it is now two years after the great impact of the pandemic when all types of methodologies should be evaluated, and in this case blended-learning deserves special interest.
4. What specific improvements should the authors consider regarding the methodology? What further controls should be considered?
Regarding the methodology, the statistical tests performed should be improved, as well as the presentation of the results, since there are figures that are not correctly displayed, for example in Figure 1, in lines 72-73, all the figures should be revised.
5. Are the conclusions consistent with the evidence and arguments presented and do they address the main question posed?
The conclusions drawn from the results are consistent, but a not too long paragraph should be added on the possibility of carrying out surveillance of these processes in the next 5 years.
6. Are the references appropriate?
Yes, the references are appropriate and fit correctly to the topic of the study because of their relevance.
7. Please include any additional comments on the tables and figures.
As already mentioned in section 4, the figures should be revised because they present problems for their correct visualization.
Reviewer 2 Report
It is necessary to increase the number of cases; perhaps since there were two cases, it would have been interesting to know more details about each of the participants in the research. For example, their level of proficiency in oral and written Spanish. In my opinion, the level of linguistic competence and digital literacy are variables that are interrelated. Regarding digital literacy, more data on teachers' prior digital training are lacking.
The subject matter is certainly of great interest. I encourage you to continue exploring this line of work.
Reviewer 3 Report
line 78. Palmer (1993) mentioned that identity is the nexus that the force that constituted his or her life converged. (this sentence is contains little sense).
line 92 The third feather emphasizes ... (Are the authors describing a poor bird with several feathers?)
line 193 what is Fundamental Spanish I remotely? (Is it a number or a preposition I? Spanish Major?)
line 228 extracted from a coursed named Fundamental Spanish One. (a new word "a coursed"?)
lines 271-275 are direct speech, thus punctuation marks are necessary.
lines 313-317 the same problem
line 325 which led to when they were asked to elaborate on an ideal... (we could not understand what it led to)
line 379-381 direct speech. Needs punctuation marks
line 403 each or combination was prioritized by oneself based (so, these combinations were not prioritized by William, but an unknown person?)
line 411 M.A in Spanish (a period is necessary M.A.)
lines 419-424 direct speech. Needs punctuation marks
line 482 An close examination (reconsider the article)
line 484 an mentorship (reconsider the article)
lines 482-484 (sentence grammatically incorrect and senatically needs reconsidering)
line 503 Therefore, despite of desiring to implement innovation, the teaching practices on the virtual platforms were no doubt display limitations. (sentence needs reconsidering)
lines 570-572 Therefore, based on the evidence retrieved from Spanish teachers’ digital identity and its formation process, the primary task to improve language teachers’ digital literacy, including workshops on contemporary education technology, the traits of learners in the online class, or multi-modal assessment. (the second part of the sentence lacks a verb)
lines 585-596 Language teachers, regardless of unmotivated or confident (what?) in online teaching,
line 587 might led (the verb is not correct)
Reviewer 4 Report
Dear author,
Congrats!
My only question is how reliable or replicable are the results considering that your corpus of analysis is based on such a small number of two?
You should mention this in the limitations.
Reviewer 5 Report
The paper addresses the issue of Digital Identity as a necessity that occured through the recent pandemic crisis considering the example of two Chineese Spanish Teachers. The authors provide a very good coverage of the state of the art, however, the results presented by the analysis have certain limitations which further limit the ability to derive concrete conclusions. In particular the analysis primarily focuses on the practices that the two teachers have adopted and the difficulties they faced from the forced transition towards the online teaching sessions. However, to derive wider conclusions the study could be benefited from at least a set of common questions that both teachers could respond to as well as a short survey on the students that have followed these sessions. Such an approach could help in mapping attributes of the digital identity taking the view of the teacher and the student (in an online session the ability as the two teachers recognised it is difficult to assess the student attitude and engagement).
Moreover, the paper is lacking of best practice information on online learning so as to compare it with the approach followed by the two teachers and allow for conclusions that could be more generic. In particular, as teachers would become more mature in adopting best practices in online sessions their digital identity would also mature towards a set of attributes.
For that reason I consider that the paper requires significant improvement in order to become publishable.
Round 2
Reviewer 3 Report
The authors showed devotion and commitment and have done their best to bring the manuscript to the state of an end product.
As some good and friendly advice: please check and double check any manuscript aginst any errors with a good translator.
Author Response
Thank you very much for your feedback! I have scrutinized the text again and edited the grammar accordingly. Please see the updated manuscript.

Reviewer 5 Report
The revised version of the paper has been improved and covers the originally provided comments.
Author Response

(The authors gave the same response as above.)
